# Gut Microbiota and Cardiovascular Disease: Evidence on the Metabolic and Inflammatory Background of a Complex Relationship

**DOI:** 10.3390/ijms24109087

**Published:** 2023-05-22

**Authors:** Antonio Nesci, Claudia Carnuccio, Vittorio Ruggieri, Alessia D’Alessandro, Angela Di Giorgio, Luca Santoro, Antonio Gasbarrini, Angelo Santoliquido, Francesca Romana Ponziani

**Affiliations:** 1Angiology and Noninvasive Vascular Diagnostics Unit, Department of Cardiovascular Sciences, Fondazione Policlinico Universitario Agostino Gemelli IRCCS, 00168 Rome, Italy; claudia.carnuccio@gmail.com (C.C.); vit_ruggieri@yahoo.it (V.R.); ale89dale@gmail.com (A.D.); angela.digiorgio@policlinicogemelli.it (A.D.G.); luca.santoro@policlinicogemelli.it (L.S.); angelo.santoliquido@unicatt.it (A.S.); 2Digestive Disease Center (CEMAD), Fondazione Policlinico Universitario Agostino Gemelli IRCCS, 00168 Rome, Italy; antonio.gasbarrini@unicatt.it (A.G.); francesca.ponziani@gmail.com (F.R.P.); 3Department of Translational Medicine and Surgery, Catholic University of the Sacred Heart, 00168 Rome, Italy

**Keywords:** gut microbiota, cardiovascular disease, systemic inflammation, dysbiosis, atherosclerosis, non-alcoholic fatty liver disease, inflammatory bowel disease, primary biliary cholangitis

## Abstract

Several studies in recent years have demonstrated that gut microbiota–host interactions play an important role in human health and disease, including inflammatory and cardiovascular diseases. Dysbiosis has been linked to not only well-known inflammatory diseases, such as inflammatory bowel diseases, rheumatoid arthritis, and systemic lupus erythematous, but also to cardiovascular risk factors, such as atherosclerosis, hypertension, heart failure, chronic kidney disease, obesity, and type 2 diabetes mellitus. The ways the microbiota is involved in modulating cardiovascular risk are multiple and not only related to inflammatory mechanisms. Indeed, human and the gut microbiome cooperate as a metabolically active superorganism, and this affects host physiology through metabolic pathways. In turn, congestion of the splanchnic circulation associated with heart failure, edema of the intestinal wall, and altered function and permeability of the intestinal barrier result in the translocation of bacteria and their products into the systemic circulation, further enhancing the pro-inflammatory conditions underlying cardiovascular disorders. The aim of the present review is to describe the complex interplay between gut microbiota, its metabolites, and the development and evolution of cardiovascular diseases. We also discuss the possible interventions intended to modulate the gut microbiota to reduce cardiovascular risk.

## 1. Introduction

Cardiovascular disease (CVD) is a leading cause of death and morbidity worldwide. Identification of possible preventive strategies is of central interest to avoid CVD onset and progression. In this regard, intensive medical and scientific activity is currently employed in the management of traditional risk factors, but despite maximum medical therapy, there is still a residual risk of undetermined etiology. The gut microbiota is gaining increasing interest as one of the potentially modifiable factors that are involved in the pathogenesis of several diseases, including CVD. A multitude of microorganisms lives symbiotically with the human host; 10–100 trillion microbes among bacteria, viruses, fungi, and helminths are located in the gut [1]. They serve a multitude of functions, which include maintenance of intestinal homeostasis and defence against external aggressive agents, modulation of the immune response, and production of metabolites. Being a living entity, the gut microbiota evolves during growth and changes under external environmental influences; owing to the lifestyle of the last 70 years, processed food intake and the increased use of antibiotics or other drugs has resulted in a modification of gut microbiota composition and diversity [2]. Furthermore, the gut microbiota is affected by changes of the host organism, such as pathological conditions that may contribute to generate a quantitative and qualitative imbalance of the bacterial communities called “dysbiosis”. Dysbiosis can be transient, with the recovery of a new steady state, or persistent, being the background for the development of chronic diseases. Indeed, in recent years, several studies demonstrated the role of gut microbiota and its metabolites in determining the onset and progression of cardiovascular and non-cardiovascular pathologies of the host (e.g., inflammatory bowel disease, colon cancer, hypertension, heart failure, and stroke). Many of these products are specific metabolic byproducts of certain bacterial species, and the most relevant are short-chain fatty acids (SCFAs), trimethylamine (TMA), bile acids (BAs), coprostanol, phenylacetylglutamine as well as lipopolysaccharide (LPS); their implication in CVD is of recent interest [3]. Thus, identifying a gut microbiota signature and its related metabolites can be useful for intervening on an undervalued novel cardiovascular risk factor. The objective of this review is to explore the relationship between gut microbiota and cardiovascular risk and CVD, and the possible intervention strategies for its modulation, prevention, and treatment.

## 2. Relationship between Gut Microbiota and Cardiovascular Risk Factors

The gut microbiota can influence and be affected by virtually all cardiovascular risk factors known to date.

The gut microbiota of patients with arterial hypertension shows a lower diversity, and an increased abundance of *Clostridiales* and *Bacterodiales* has been observed in men and mice models of hypertension [4]. In the Coronary Artery Risk Development in Young Adults (CARDIA) study, the abundance of the *Robinsoniella* was negatively associated with systolic blood pressure [5]. Germ-free mice were protected from angiotensin-II-induced hypertension and from angiotensin-II-induced cardiac inflammation and remodeling [6]; furthermore, when compared with chronic angiotensin-II infusion mice models or healthy controls, spontaneous hypertensive mice have a high *Firmicutes* to *Bacteroidetes* ratio [7]. This suggests a pathogenic mechanism linking arterial hypertension and the gut microbiota. Cigarette smoke can directly and indirectly alter the gastrointestinal barrier. It can up-regulate enzymes involved in oxidative stress damage. The gut microbiota has a different composition in smokers than in non-smokers and resembles that of patients affected by inflammatory bowel diseases; in particular, an increased relative abundance of *Actinobacteria* and *Cyanobacteria* has been reported in smokers than in non-smokers. There is also limited evidence on the impact of tobacco cessation on the gut microbiota, which seems mainly to produce an increase in *Firmicutes* and a reduction in *Bacteroidetes*. However, whether these changes may have an impact on cardiovascular risk is still unclear [8].

Furthermore, the gut microbiota can alter the plasma lipoprotein profile both by reducing cholesterol biosynthesis [9] and plasma cholesterolemia through different mechanisms (i.e., incorporation into bacterial cell membranes, deconjugation of primary bile acids into secondary bile acids, conversion into coprostanol which is eliminated in the feces, production of SCFAs) [10]. Some bacterial products have shown specific effects on lipid metabolism; for example, exopolysaccharides (EPSs) derived by *Agaricus brasiliensis* have a cholesterol-lowering effect in mice [11]. Gut microbiota can also interact with dietary lipids, producing active compounds with a regulatory effect on plasma lipoproteins. Among these compounds, conjugated linoleic acid is able to lower cholesterol, triglyceride, and lipoprotein levels in vivo and in vitro studies [12]. 

## 3. Pathogenic Role of the Gut Microbiota in CVD: Focus on Metabolites and Inflammation

The gut microbiota is a metabolically active superorganism. Its products complement the metabolic functions of the host and can influence human health. In addition, the gut microbiota has an enormous immunological potential, being capable of triggering a systemic inflammatory response with consequences also for the cardiovascular system (Figure 1).

### 3.1. Trimethylamine N-Oxide

Trimethylamine N-oxide (TMAO) is an endothelial toxic factor produced by the gut microbiota from ingested foods [13]. Indeed, meat, egg yolk, and high-fat dairy products normally contain choline, phosphatidylcholine, and carnitine-trimethylamine that are subjected to a two-step modification process: (1) are metabolized by the gut microbiota into TMA, which is absorbed and reaches the liver through the portal circulation; (2) in the liver, TMA is oxidized to TMAO by flavin-monooxygenase-3 (FMO3), and then re-enters the systemic circulation. Although, the precise mechanism through which TMAO enhances cardiovascular risk is not fully understood, TMAO plays a critical role in endothelial cell dysfunction and atherosclerotic plaque formation, mainly promoting inflammation and stimulating platelet reactivity through extracellular signal-regulated protein kinases 1 and 2 (ERK1/2) and Jun N-terminal kinase (JNK) phosphorylation [14,15]. An in vitro study showed that in endothelial and smooth muscle cells, TMAO induces the expression of genes involved in the inflammatory response, such as cyclooxygenase 2 and interleukin (interleukin)-6, and enhances the adhesion of leukocytes to the vascular endothelium causing a vascular endothelial inflammatory damage [16]. Furthermore, TMAO promotes foam cells formation, which are the main component of atherosclerotic plaques, by stimulating the adhesion of monocytes to the vascular endothelium and up-regulating scavenger receptors on the macrophage membrane [17]. In humans, TMAO plasma concentration ranges from 0.09 to 141.2 μM (mean 6.96 μM, median 3.07 μM); high levels are an independent risk factor for CVD and correlate with incidence of major adverse cardiac events (MACE) [18]. A recent study including 956 subjects, 471 of whom with stable chronic heart failure, reported that high serum levels of trimethyllysine (TML), a TMAO precursor, were associated with a higher risk of CV death, all-cause mortality, and re-hospitalization [19]. Another recent clinical study demonstrated that high levels of TMAO were predictive of a 4-fold increase in the risk of mortality for all causes in patients with stable coronary artery disease and angiographic evidence of significant stenosis (COURAGE-like patient cohort). The 5-year risk of mortality was independent of traditional risk factors, high-sensitivity C-reactive protein, and estimated glomerular filtration rate, thus confirming the important prognostic role of TMAO [20]. The PEGASUS-TIMI 54 trial also confirmed the prognostic role of TMAO in stable patients with prior myocardial infarction, emphasizing that high levels of TMAO were associated with cardiovascular death and stroke, but not with recurrent myocardial infarction. Furthermore, TMAO induced cardiac inflammation and fibrosis, leading to cardiac dysfunction in animal models. A study also reported a high rate of apoptosis and necrotic cell death after myocardial infarction following TMAO or a high choline diet in myocardial infarction mice and in primary cardiac fibroblasts cultures. This is probably due to different mechanisms, such as an increased transformation of fibroblasts into myofibroblasts and by enhancing inflammation and promoting cardiomyocytes death [21].

### 3.2. Short-Chain Fatty Acids

Acetate, butyrate, and propionate are metabolites that derive from the intestinal bacteria fermentation of indigestible polysaccharides and are classified as SCFAs. SCFAs exert beneficial effects not only in the gut, promoting intestinal barrier integrity by enhancing the expression of proteins of the cellular junction and serving as energy substrates for epithelial cells, but also on metabolic functions and inflammation. Indeed, they can modulate glycemic control and lipid metabolism, exhibit anti-inflammatory and anti-tumorigenic activity, decrease oxidative stress, and modulate the secretion of inflammatory cytokines and chemokines [22]. Some data suggest that SCFAs may have protective effects on atherosclerosis and CVD. Propionate moderately reduces blood pressure through vasodilation, mediated by the activation of G-protein-coupled receptors 41 (Gpr41), localized on the vascular endothelium; the chronic reduction in blood pressure could be mediated by the improvement of endothelial dysfunction [23]. Moreover, propionate has been shown to exert cardioprotective properties, being associated with reduced cardiac remodeling in hypertensive subjects. In hypertensive left ventricular hypertrophy, there is a different spatial redistribution of gap junctions, which causes a potential risk of pathological electrical activity. It has been shown that oral propionate supplementation in the animal model can reduce electric remodeling by a reduced lateralization of connexin 43 in cardiomyocytes, thus reducing susceptibility to ventricular tachycardia in vivo [24]. An intriguing relationship between SCFAs and blood pressure has also been described. In fact, some human studies have found that a lower blood pressure was associated with SCFA-producing gut microbiota. A murine study has suggested that hypertension is related to reduced blood levels but high fecal concentrations of butyrate, possibly because of dysfunctional butyrate intestinal transporter [25]. Histone deacetylases (HDAC) activation has been associated with arterial hypertension; in fact, in animal models, its long-term inhibition mediated by valproic acid could attenuate not only the increased inflammatory response, but also mean arterial pressure and cardiac remodeling and hypertrophy [26]. SCFAs could affect blood pressure through HDAC inhibition, acting directly on vascular and renal receptors, but also through the gut–brain axis via afferent enteric and colonic vagus nerve signaling [27,28,29]. Nevertheless, butyrate is a negative modulator of inflammation, and this anti-inflammatory activity is mediated by inhibition of HDAC, which normally regulates innate immunity pathways, controlling myeloid cell differentiation and inflammatory response mediated by toll-like receptors (TLR)- and interferon (IFN)- inducible gene expression [30,31]. Thus butyrate, via HDAC inhibition, suppresses the production of pro-inflammatory cytokines, such as tumor necrosis factor-α (TNF-α), IL-12, and interferon-γ (IFN-γ), enhancing the production of IL-10 by monocytes in vitro, which exerts anti-inflammatory properties [32]. Finally, SCFAs are also involved in the intestinal metabolism of cholesterol. An inverse correlation has been found between serum levels of cholesterol and the conversion of cholesterol to coprostanol in patients with high SCFAs fecal concentrations; this may be due to a different composition of the gut microbiota, although the exact mechanisms have yet to be fully clarified [33]. 

### 3.3. Bile Acids

Bile acids (BAs) are involved in metabolic disease, inflammatory bowel disease, and gastrointestinal carcinoma, but also in CVD. For example, cardiac toxicity of BAs is known since the 1960s, and to date, we learned that BAs can cause cardiac remodeling and electrophysiological alterations, predisposing to fatal arrhythmic events. BAs interact with nuclear receptors, such as the farnesoid-X receptor (FXR) and vitamin D receptor, which are also expressed in the heart. In cardiomyocytes, FXR activates mitochondrial permeability transition pore (MPTP), a protein involved in starting the cellular apoptotic process and linked to heart failure. The relationship between BAs and heart disease is best highlighted by the two main cholestatic disorders in humans: primary biliary cholangitis (PBC) and primary sclerosing cholangitis (PSC), which are associated with impaired cardiovascular function. In particular, PBC patients have a prolonged QTc interval and an abnormal left ventricular (LV) ejection time in response to the assumption of upright position to clinostatism; these changes have been identified as possible consequences of an impaired BAs metabolism [34]. Another recent study showed that circulating BAs levels are predictive of coronary heart disease (CAD) in humans [35]. Moreover, BAs may contribute to endothelial dysfunction and atherosclerosis. Indeed, BAs enhance the expression of intracellular adhesion molecule-1 (ICAM-1) and vascular adhesion molecule-1 (VCAM-1), through an interaction with FXR, commonly expressed at the level of human endothelial cells, so contributing to the leucocyte-induced inflammation in vascular tissues. However, FXR stimulation may have other metabolic and vascular effects such as decreasing inflammatory gene expression in macrophages and blocking the proliferation, migration, and activation of vascular smooth-muscle cells [36].

### 3.4. Coprostanol

Coprostanol is a non-absorbable sterol produced in the gut by the conversion of cholesterol and eliminated with feces. The conversion of cholesterol to coprostanol has a biologically relevant effect, because it is associated with a reduction in blood cholesterol levels [37]. However, the amount of coprostanol produced in humans is extremely variable, with a predominance of high converters over low or inefficient converters. [38]. Intestinal bacteria such as *Eubacterium coprostanoligenes*, *Bacteroides* spp., *Lactobacillus* spp., and *Bifidobacterium* spp. are able to convert cholesterol into coprostanol. A cholesterol-inducible enzyme named intestinal steroid metabolism A (IsmA) has been recently discovered and is involved in the oxidation of cholesterol to 4-cholesten-3-one and of coprostanol to coprostanone. Interestingly, this enzyme is overexpressed in *E. coli* and the presence of IsmA genes in other microbial species has been associated with a reduction in blood cholesterol levels [39]. Sekimoto et al. described that an increased coprostanol/cholesterol ratio in the stool is inversely proportional to blood cholesterol levels [40]. The study of coprostanoligenic strains may be of clinical interest in order to reduce cardiovascular risk by modulating the microbiota [10].

### 3.5. Phenylacetylglutamine

Phenylacetylglutamine (PAGln) is a metabolite derived from the conjugation of glutamine and phenylacetate operated by the gut microbiota. PAGln has been associated with increased platelet activity and thrombosis potential [41]; recent studies point out that PAGln serum levels are correlated with coronary atherosclerosis severity and the incidence of CVD and MACE. Another study shows that high levels of PAGln are independently associated with an increased risk of coronary in-stent restenosis, which is a factor related to a worse prognosis in patients with CAD [42].

### 3.6. Vitamin K2

Vitamin K2 or menaquinone (MK) is a vitamin K isoform and a cofactor involved in the carboxylation of various proteins [43]. Among these, some are involved in maintaining the structural and functional integrity of the vascular wall, such as the Matrix Gla-protein (MGP). MGP binds calcium crystals and inhibits the pro-mineralizing factor bone morphogenetic protein-2 (BMP-2), preventing arterial calcifications [44]. Quantitative changes in gut microbiota composition, as in the case of small intestinal bacterial overgrowth (SIBO), are associated with alterations in vitamin K2 metabolism. In particular, serum levels of dephosphorylated-uncarboxylated matrix Gla-protein, the inactive form of MGP, are significantly higher in patients with SIBO compared with controls, independently to vitamin K2 daily intake. Patients with higher levels of inactive MGP showed early signs of vascular dysfunction, documented by an increase in arterial stiffness measured by pulse-wave velocity [45]. 

### 3.7. Gut Microbiota-Derived Inflammation

Inflammation is associated with the development of cardiovascular disease [46]. Systemic inflammation is strictly connected with accelerated atherosclerosis [47]. The gut microbiota may also influence cardiovascular risk through a pro-inflammatory effect, exerted not only by its metabolites but also by bacteria themselves, especially under conditions of dysbiosis [48]. LPS, a part of the outer membrane of Gram-negative bacteria that is released in the systemic circulation at the time of cell death, is a well-known trigger of systemic inflammation in conditions of increased translocation of intestinal bacteria or their products in the bloodstream [49]. Scientific evidence over the past twenty years supports a contribution of LPS in the development of atherosclerosis, especially as regards to foam cells formation and cholesteryl ester accumulation from native low-density lipoproteins. Furthermore, through its pro-inflammatory effects, LPS promotes the secretion of TML, a TMAO precursor, and other pro-inflammatory cytokines from monocytes-macrophages. LPS is able to bind TLRs, activating a series of immune reactions. In particular, in the endothelium of blood vessels, LPS binds TLR4, which activates myeloid differentiation primary response 88 (MYD88) and nuclear factor kappa B (NFκB) pathways, leading to an enhanced synthesis of IL-6, IL-1, IL-27, and TNF-α, with pro-inflammatory effects. Increased LPS serum levels have been documented in subjects with coronary artery disease, supporting a possible role as a disease marker [50]. The correlation between gut microbiota and inflammation also occurs through platelet hyperactivation mediated by TMAO. In particular, it has been proven that plasma rich in TMAO enhances human platelet reactivity not specifically linked to a specific stimulus, rather to multiple agonists, including adenosine diphosphate (ADP), thrombin, collagen, and arachidonic acid. In fact, TMAO interacts with the phospholipids of platelet membrane promoting cellular activation in response to sub-maximal agonist stimulation; this effect is also mediated by the enhancement of agonist-dependent release of Ca^2+^ from intracellular platelet stores and increase of the inositol-1,4,5-trisphosphate (IP3)-related signaling [51]. Activated platelets also release CD40 ligand (CD40L) and other mediators, triggering an inflammatory response in the endothelium, which results in endothelial dysfunction [52]. The gut microbiota also has a strong modulating effect on the immune response [53]. Nod-Like Receptor Protein 3 (NLRP3) inflammasome is a multiprotein complex belonging to the innate immune system, which coordinates the activation of the inflammatory response triggered by a variety of stimuli, such as pathogens and their products (the so-called PAMPs, Pathogen-Associated Molecular Patterns) and molecules released from damaged enterocytes (the so-called DAMPs, Damage-Associated Molecular Patterns). The NLRP3 inflammasome can undergo two different types of activation, the canonical and the non-canonical one. In the canonical inflammasome pathway, a first signal mediated by microbial components and cytokines, i.e., ligands for toll-like receptors (TLRs) or cytokine receptors, primes the NLRP3 inflammasome to activate the transcription factor NF-κB, which upregulates the expression of NLRP3 and pro-IL-1β (priming phase). The NLRP3 inflammasome is then activated by bacterial and fungal toxins, viral RNA, extracellular matrix components, ATP, ionic flux (in particular potassium efflux), reactive oxygen species (ROS), and mitochondrial dysfunction (activation phase). In this phase, NLRP3 oligomerizes with an ASC, i.e., apoptosis-associated speck-like protein containing a caspase recruitment domain (CARD), and induces pro-caspase-1 activation in caspase-1, as well as IL-18 and IL-1β release. It has been shown in a murine model that NLRP3 inflammasome deficiency in mice may alter gut microbiota composition by increasing mucosal bacteria and favoring systemic bacterial translocation [54]. Moreover, increased levels of IL-1β and IL-18 by inappropriate activation of the NLRP3 inflammasome due to cholesterol crystals and oxidized low-density lipoprotein (oxLDL) have been shown to be leading mechanisms in the pathogenesis of atherosclerosis [55,56,57,58,59]. The non-canonical pathway is triggered by the intracellular activation of caspase-11 in mice and caspases-4 and 5 in humans by LPS, without TLR interaction. Caspase-11 directly activates the NLRP3 inflammasome, inducing pyroptosis through the cleavage of gasdermin D and release of IL-1β and IL-18 [60]. In a murine study, it has been shown that caspase-11 has a protective function against intestinal inflammation, probably via the release of IL-18; in fact, caspase-11-/- mice are more susceptible to dextran sodium sulphate (DSS)-induced colitis, and their gut microbiota is characterized by a markedly reduced prevalence of the phylum *Prevotella* [61].

## 4. Gut Microbiota Composition in CVD: What We Know So Far

### 4.1. Atherosclerosis and Coronary Artery Disease

The close correlation between atherosclerosis and gut microbiota has been extensively described in literature. Dysbiosis and microbial metabolites (i.e., TMAO) play a role in the pathogenic mechanisms of atherosclerosis, such as systemic inflammation, endothelial dysfunction, and lipid homeostasis, and are associated with the severity of the disease. The increase in *Enterobacteriaceae* is associated with larger coronary plaque fibrotic area and more severe coronary atherosclerosis [62]. Recent studies have shown a correlation between specific intestinal bacteria (i.e., *Dysgonomonas*, *Paraprevotella*, *Succinatimonas*, and *Bacillus*) and plaque vulnerability, intended as the presence of a thin-cap fibroatheroma, lipid-rich plaque with necrotic core, macrophages, microvessels, cholesterol crystals, and large plaque burden [63]. Moreover, marked overexpression of TMA-producing intestinal microbial enzymes was observed in patients with coronary artery disease compared with healthy controls [64].

### 4.2. Heart Failure

The gut microbiota of patients with chronic heart failure is characterized by a decreased abundance of beneficial butyrate-producing bacteria and an increase in pathogenic bacteria, including *Campylobacter*, *Salmonella*, *Shigella*, *Yersinia enterocolitica,* and *Candida* species [1]. Patients with chronic heart failure and cardiac cachexia also show intestinal wall congestion and oedema, impaired microcirculation, and increased intestinal permeability; this results in gut dysbiosis with a predominance of *Firmicutes*, *Bacteroidetes,* and *Proteobacteria*, as well as in the translocation of bacteria and their metabolites with potential effects on cardiovascular health [65]. 

### 4.3. Stroke

It has been observed that a gut microbiota enriched with SCFAs-producing bacteria, such as *Akkermansia*, *Victivallis*, *Ruminococcaceae*, and *Odoribacter*, may lead to increased risk of cerebrovascular events and correlates with their severity. Conversely, stroke has been linked to gut dysbiosis and intestinal barrier dysfunction [66]. After stroke, a lower blood supply leads to ischemic intestinal damage and this results in the production of excessive nitrate through free radical reactions, but also in changes in the gut microbial community, with the increased abundance of *Enterobacteriaceae*. A recent animal study investigated the association between acute ischemic stroke and gut dysbiosis, showing how *Enterobacteriaceae* represent a biomarker of primary poor outcome; functionally, this is explained by increased systemic inflammation, worsening brain ischemia. The authors also highlighted that the administration of aminoguanidine or superoxide dismutase could be useful in counteracting brain injury by restoring gut dysbiosis [67]. Moreover, after stroke, the production of neurotransmitters is altered and the release of noradrenalin leads to changes in intestinal permeability, causing the activation of corticotropin-releasing hormones and glucocorticoid hormones, and promotes bacterial translocation [68]. Finally, TMAO promotes endothelial dysfunction, vascular inflammation, and changes in small cerebral arteries including lipohyalinosis, fibrinoid necrosis, and microaneurysm formation, thus favoring the onset of hemorrhagic stroke. In a Chinese study involving 622 patients with a first stroke, a significant correlation was found between the TMAO serum level and first hemorrhagic stroke [69].

## 5. Dysbiosis as a Condition Predisposing to CVD

Many inflammatory and metabolic diseases associated with gut dysbiosis also have a close association with CVD.

Several meta-analyses of cohort studies have reported associations between inflammatory bowel diseases (IBDs) and CVD [70,71]. In particular, increased carotid intimal thickness, wall stiffness, and endothelial dysfunction [72,73], a 4-to-5 fold increase in homocysteine serum level [74], and a higher prevalence of acute myocardial infarction (AMI) have been observed in patients with IBD compared with those without inflammatory intestinal diseases, with the highest risk of AMI in young women aged 30–34 years [75]. A Danish cohort study has also reported that patients affected by IBD have an increased risk of stroke [76,77], as well as a higher rate of atrial fibrillation and hospitalization for heart failure, even in young patients, and especially during acute flares or in case of persistently active disease [78,79]. It is well-established that IBDs are associated with gut dysbiosis [80,81] and several studies have shown a generalized decrease in microbial diversity and a reduction in specific beneficial bacterial taxa, including *Lactobacillus*, *Eubacterium,* and butyrate-producing bacteria such as *Faecalibacterium prausnitzii* [82,83,84,85,86,87,88]. Non-alcoholic fatty liver disease (NAFLD), which can be considered the hepatic manifestation of metabolic syndrome, also finds its roots in dysbiosis and increased intestinal permeability [89]. Several studies have identified a reduced gut microbiota diversity and various changes in gut microbiota composition in patients with NAFLD [90,91,92]. Dysbiosis, in turn, is responsible for the dysregulation of intestinal endothelial and vascular barrier function, with enhanced translocation of bacteria and their products (PAMPs: endotoxins, LPS, peptidoglycan) and molecules released from damaged intestinal cells (DAMPs); after reaching the liver through the portal circulation or the bloodstream through mesenteric lymph nodes, PAMPs and DAMPs trigger various cellular signaling pathways that induce a systemic inflammatory response [93,94]. Alterations in the gut microbiota were found in patients with coronary artery disease and NAFLD, mainly characterized by a decrease in *Parabacterioides* and *Colinsella* [95]. A prospective study has also shown that TMAO concentrations were higher in patients with NAFLD, being significantly and independently associated with an increased risk of all-cause mortality [96]. Intriguingly, this association was not present in subjects without NAFLD. This confirms that altered gut dysbiosis may influence progression of metabolic syndrome-associated diseases as NAFLD [97]. Another interesting setting are patients with PBC, an autoimmune cholestatic liver disease that is characterized by hypercholesterolemia and affects middle-aged women. However, there is limited data on the incidence of atherosclerosis and CVD in these patients, with only recent studies showing an increased risk [98]. Patients with PBC have a doubled prevalence of lower extremity arterial disease (LEAD) compared with age-matched general female population, and the gut microbiota seems to be associated with this finding [99]. Indeed, vascular adhesion molecule-1 (VCAM-1) and TNF-α were independent predictors of LEAD, and *Acidaminococcus*, a bacterial genus highly abundant in PBC women, was positively correlated with serum levels of TNF-α. A final remark should be made on rheumatic diseases such as rheumatoid arthritis (RA) and systemic lupus erythematous (SLE). RA has been associated with periodontitis [100], and high serum titers of *Porphyromonas gingivalis* antibodies have been found in patients with more severe disease activity and functional impairment [101,102]. Interestingly, *P. gingivalis* DNA was detected also in atherosclerotic plaques of subjects with periodontitis. Clinical and animal studies have pointed out that *P. gingivalis* accelerates atherosclerosis [103,104]; the mechanisms can be multiple, and include: (a) intracellular influx of oxidized-LDL and its conversion to cholesterol crystals, via increased expression of CD36 and fatty acid binding protein 4 (FABP4) on macrophages [105,106]; (b) activation of NLRP3 inflammasomes by cholesterol crystals’ damage to the phagolysosomes and *P. gingivalis*-induced production of reactive oxygen species, with consequent activation of the inflammatory cascade [107,108]; (c) down-regulation of the cholesterol transporters ATP-binding cassettes (ABCA1) on macrophages, which promotes cholesterol accumulation [109]. The pathogenesis of SLE has been associated with dysbiosis, which is mainly characterized by a lower *Firmicutes/Bacteroides* ratio and overabundance of *Ruminococcus gnavus*, *Enterococcus gallinarum*, *Streptococcus anginosus*, *Streptococcus dispar*, *Veillonella*, and *Campylobacter*, and contributes to disease development and progression through pro-inflammatory stimulation and production of anti-dsDNA antibodies [110,111,112]. The pro-inflammatory milieu interferes with blood pressure regulatory mechanisms, such as the renin–angiotensin system and the sympathetic nervous system [113]; in particular, circulating TNF-α, which is increased in patients with SLE and correlates with disease activity, is involved in the development of hypertension; indeed, in female murine models of SLE, a decrease in mean arterial pressure has been observed after treatment with the anti-TNF-α antibody etanercept [114,115]. Furthermore, LPS increases the expression of TLR4 in blood vessels, which results in increased nicotinamide adenine dinucleotide phosphate (NADPH) oxidase-dependent superoxide production, inflammation, and endothelial dysfunction [116,117]. Elevated levels of plasma LPS have been reported in both patients and hypertensive mice affected by SLE [118,119]. Lastly, in patients with SLE, intestinal dysbiosis is associated with an altered production of SCFAs [120].

In summary, dysbiosis is involved in the pathogenesis of several diseases that are also associated with cardiovascular dysfunction. Microbial-derived inflammation but also metabolites are the driving force of this link, but it is not possible to exclude further mechanisms that have not been clarified to date.

## 6. Evidence on the Impact of Gut Microbiota Modulation in Reducing Cardiovascular Risk

The recognition of the gut microbiota as a key player in the pathogenesis of CVD suggests that specific therapeutic interventions aimed at its modulation may potentially reduce cardiovascular risk. Several studies have focused on this topic; however, they have shown mixed results.

### 6.1. Dietary Intervention

Diet has multiple effects on the gut microbiota, being able to modify its composition and function [121]; these effects are not immediate and have greater impact if maintained for a long period [122]. A diet rich in plant products modulates the intestinal bacteria community favoring the growth of species able to ferment fibers, resulting in increased production of SCFAs and phosphatidylcholine. Conversely, a high-fat diet leads to unfavorable changes in the gut microbiota, fecal metabolomic profile, and systemic inflammation [123]. All of these changes are associated with adverse effects on human health, because in the long term they lead to increase the risk of obesity, metabolic syndrome, and cardiovascular risk [124,125]. A controlled-feeding trial conducted in China on 217 healthy volunteers highlighted that a high-fat diet modifies the gut microbiota composition, increasing *Bacteroides* and *Alistipes*, more abundant in patients with type 2 diabetes mellitus (T2DM), and reducing *Faecalibacterium*, a butyrate-producing microorganism. Conversely, in the low-fat diet group, an increased abundance of *Faecalibacterium* and *Blautia* was found, which favorably affect lipid metabolism [126]. Changes in the gut microbiota community reflect in fecal metabolomic profiles. In fact, a high-fat diet leads to a reduction in SCFAs and an increase in arachidonic acid and LPS biosynthesis, with consequent elevation in circulating pro-inflammatory factors (i.e., plasminogen activator inhibitor-1, IL-1, and TNF-α mRNA) [127]. In a randomized, controlled, crossover study conducted in subjects with ischemic heart disease, a 4-week consumption of a diet rich in plant-based products (vegetarian diet), has been shown to reduce oxidized low-density lipoprotein cholesterol and to change the relative abundance of intestinal bacteria, in particular of *Ruminococcaceae* and *Barnesiella*, and their metabolites compared with Mediterranean diet. An improvement in cardiovascular risk profile derived from reduction in oxidized LDL cholesterol was reported in the vegetarian diet group, but no significant difference in TMAO production was observed [128]. Another study showed that a Mediterranean diet has anti-inflammatory properties, with a negative correlation between SCFAs production and the expression of inflammatory cytokines such as vascular-endothelial growth factor (VEGF), monocyte chemoattractant protein-1 (MCP-1), IFN-γ-induced protein 10 (IP-10), IL-17, and IL-12. In addition, Mediterranean diet increased the abundance of *Enterorhabdus*, *Lachnoclostridium,* and *Parabacteroides* [129]. Another observational study conducted by Dong D. Wang et al. supports the hypothesis that a controlled diet may produce beneficial changes in the gut microbiota composition in prevention of CVD [130]. Long-term adherence to a Mediterranean diet exerts a gradual selective pressure on the adult gut microbiota, resulting in a relative abundance of fiber-metabolizing bacterial species (*Faecalibacterium prausnitzii*, *Bacteroides cellulosilyticus,* and *Eubacterium eligens*), compared to other pathogens mainly associated with a Western-type diet and red meat intake (such as *Ruminococcus torques*, *Clostridium leptum,* and *Collinsella aerofaciens*). For example, the selection of particular taxa responsible for the conversion of BAs (i.e., *C. aerofaciens*) may lead to adverse cardiometabolic effects by a dysregulation of the BAs pool; in fact, such compounds behave as hormones that interact with nuclear and G protein-coupled receptors interfering with certain metabolic processes. Other authors also refer to *Prevotella copri*, the role of which has not yet been sufficiently clarified, but seems to be associated with an increase in the biosynthesis of branched-chain amino acids, which are linked to an increased cardiovascular risk through insulin resistance in humans [131]. The adoption of a Mediterranean diet, according to Dong D. Wang et al., is able to mitigate this risk, as it is likely that such subjects do not acquire or retain *P. copri*. Another hypothesis is that the Mediterranean diet exerts its beneficial effects on CVD prevention only in *P. copri* non-carriers [130]. Many of the clinical trials currently available in literature have the major limitation of being restricted to a limited period; therefore, whether dietary changes can be maintained over time is still a matter of debate, and future studies are needed to test the resilience of the gut microbiota. The intake of particular foods within a balanced diet can produce a benefit for human health. Plant-based omega-3 fatty acids, α-linolenic acid, and polyunsaturated fatty acids (PUFAs) have been shown to exert benefits on the cardiovascular system. Notably, several of these substances can be classified as prebiotics, which are substrates selectively used by host microorganisms conferring a health benefit [132]. Walnuts are a main source of these and many other compounds, such as hydrolyzable tannins and fibers, which can be metabolized by gut bacteria and confer additional benefits in term of cardiometabolic risk. A recent randomized controlled trial involving 42 patients at cardiovascular risk (defined as overweight and obese middle-aged men and women, with dyslipidemia and hypertension) showed that a whole walnut-based diet and a walnut fatty acid–matched diet are able to change the intestinal bacterial composition, in particular increasing the abundance of *Roseburia*, a butyrate-producing bacteria. Moreover, ellagitannins naturally contained in walnuts, are metabolized by gut bacteria to form urolithins, which may provide additional cardiovascular benefit [133]. Dietary PUFAs are also associated with multiple cardiovascular effects mediated by changes in the gut microbiota; in fact, omega-3 PUFAs increase the abundance of several SCFAs-producing bacteria, decreasing those associated with TMA production. Additionally, omega-3 PUFAs would help maintain intestinal barrier integrity, thereby preventing the translocation of intestinal products into systemic circulation and reducing the production of pro-inflammatory cytokines [134]. A similar effect was also described for the polyphenol hesperidin, found in citrus fruits. Polyphenols are natural products of plant extraction potentially used as prebiotics. Dietary supplementation of polyphenols has proven to have cardiovascular benefits, because they reduce blood pressure and improve endothelial dysfunction and lipid profile [135]. Hesperidin promotes the growth of beneficial bacteria, such as *Lactobacillus* and *Bifidobacterium*, stimulates the production of SCFAs, and lowers plasma levels of pro-inflammatory cytokines such as IL-1β, TNF-α, and IL-6 [136]. Finally, there are scant results from human intervention studies with SCFAs in order to reduce blood pressure; Roshanravan et al. have demonstrated in a randomized, double-blind, placebo-controlled trial that oral butyrate supplements tend to significantly lower blood pressure in patients with metabolic syndrome [137].

### 6.2. Probiotics

The role of probiotics in reducing cardiovascular risk is an issue that is gathering scientific interest, especially in light of new scientific evidence supporting a role for the gut microbiota in the pathogenesis of CVD. A pilot study including 21 men with stable coronary artery disease showed that a 6-week daily supplementation with *Lactobacillus plantarum* 299v (Lp299v) has a favorable impact on CVD inducing changes in gut microbiome-derived circulating metabolites. Supplementation with Lp299v can improve endothelium-dependent vasodilation of the brachial artery, through the increase in nitric oxide bioavailability, and reduce systemic inflammation. These effects appear to be independent of traditional cardiovascular risk factors, and not related to TMAO serum levels [138]. Some studies show that *Bacteroides* depletion in humans is associated with higher incidence of symptomatic atherosclerosis. Nevertheless, in animal models, oral supplementation of *Bacteroides vulgatus* and *Bacteroides dorei* reduced atherosclerotic plaque inflammation and slowed its formation. In particular, *Bacteroides* supplementation in atherosclerosis-prone mice reduces LPS production, and successfully ameliorates endotoxemia, suppressing pro-inflammatory immune responses [139]. *Akkermansia muciniphila* is a component of the gut microbiota that exerts favorable effects on the pathogenesis of CVD and arterial hypertension [4]. Probiotic bacteria in milk, yogurt bacteria, and cheese starter bacteria are able to produce bioactive peptides with antihypertensive function. A study has shown that *Lactobacillus helveticus* produces angiotensin converting enzyme (ACE) inhibitory tripeptides, which play an antihypertensive role in renin-angiotensin system [140]. Studies conducted on spontaneous hypertensive rats reported that oral supplementation of high doses of *Lactobacillus casei* strain C1 led to a significant reduction in systolic and diastolic blood pressure at 8 weeks [141]. Trials conducted in humans underline that the reduction of systolic and diastolic blood pressure by probiotic supplements is modest but significant; the main limits on the effectiveness are the duration of treatment, dosage, age of the subjects, and type of strain used [142]. Probiotics may also be a therapeutic opportunity for the treatment of stroke. A recent study conducted in mice models of ischemic stroke showed that ischemic brain injury was reduced by 52% and the neurological outcome ameliorated after treatment with probiotic bacteria (such as *Bifidobacterium breve*, *Lactobacillus bulgaricus*, *Lactobacillus casei*, and *Actobacillus acidophilus*); the neuroprotective effect was probably due to the anti-inflammatory properties and modulation of oxidative stress damage [143].

### 6.3. Drugs

Many non-antibiotic medications can modulate microbiota composition and function, influencing health outcomes. For example, proton pump inhibitors are among the most widely used drugs that can modify the gut microbiota, leading to a decreased colonization-resistance to enteric infections (i.e., *Clostridium difficile* infection) and to the oralization of the colonic microbiota [144,145]. Several antidiabetic drugs, such as metformin and liraglutide, exert their therapeutic effects and additional metabolic benefits by changing the gut microbiota composition and metabolism [146]. Among the novel sodium/glucose cotransporter 2 inhibitors (SGLT2i), empagliflozin also shows an effect in increasing the richness and diversity of the gut microbiota, improving inflammatory parameters. In a recent study, empagliflozin was able to promote a selection of SCFAs-producing bacteria, such as *Roseburia*, *Faecalibacterium,* and *Eubacterium* over potentially harmful bacteria, including *Escherichia-Shigella*, *Bilophila*, and *Hungatella* [147]. In another study, a 28-day treatment with dapagliflozin, another SGLT2i, significantly improved cardiac function in the non-diabetic myocardial infarction mice model and modified the gut microbiota composition, increasing the abundance of beneficial bacteria such as *Lactobacillaceae* [148]. *Muribaculaceae* and *Lactobacillaceae* were the main components of the intestinal microbial community after treatment with dapagliflozin, while *Muribaculaceae* and *Erysipelotrichaceae* were the ones associated with myocardial infarction. Antihypertensive medications, such as the angiotensin-converting enzyme inhibitor captopril, have shown beneficial effects on hypertension-associated gut pathology, in particular reducing intestinal permeability, thickness of the muscularis layer and increasing the length of villi by 55% [149]. Pharmacological effect of cholesterol-lowering drugs is partly impaired by gut microbiota, and dysbiosis can generate further pharmacological variability [150]. In an animal model, it has been shown that the hypolipidemic effect of statins, particularly simvastatin, is partially reduced with concomitant administration of antibiotics [151]. Other studies focused on the modulating effect of statins on the gut microbiota composition; for example, in animals the administration of rosuvastatin increased the abundance of *Lachnospiraceae* and *Erysipelotrichaceae* and decreased the abundance of *Proteobacteria*, *Coriobacteriaceae* and *Akkermansia* [152]. Finally, antibiotic treatment disrupts gut microbiota homeostasis, and leads to potentially harmful dysbiosis. Rifaximin exerts anti-inflammatory and eubiotic effects, producing a positive modulation of the gut microbiota and reducing intestinal bacteria adherence, internalization, and translocation [153]; however, its effects on cardiovascular risk reduction have not yet been investigated. At present, the use of broad-spectrum antibiotics in reducing cardiovascular risk remains controversial, due to the potential side effects and the induction of bacterial resistance [1]. The concept of pharmacomicrobiomics, which means the impact of the gut microbiota on drug bioavailability, bioactivity, or toxicity by direct and indirect mechanisms, is currently emerging [154]. The role of the gut microbiota in influencing the effectiveness of a therapeutic treatment has already been investigated in various diseases such as ulcerative colitis, Crohn’s disease, and RA. Pharmacomicrobiomics is also applied in the cardiovascular field; in fact, it is known that digoxin, a drug used in heart failure, is ineffective in 1 in 10 patients because it is likely converted into an inactive form, dihydrodigoxin, by *Eggerthella lenta* [155]. In summary, pharmacomicrobiomics emphasizes the importance of pursuing a personalized medicine that focuses on the microbiota.

### 6.4. Small Molecule Antimicrobial Enzyme Therapeutics

Blocking microbial TMA production is a potential therapeutic strategy for the prevention and treatment of atherosclerosis. Selective enzymes structurally similar to choline have been developed and are being studied to reduce cardiovascular risk. In particular, 3,3-dimethyl-1-butanol (DMB) can inhibit microbial TMA-lyase and reduce both TMA production and serum levels of TMAO in mice fed a high carnitine or choline diet. DMB showed direct effects on atherosclerosis, inhibited dietary choline-dependent accumulation of cholesteryl ester in macrophages (foam cell formation) and development of aortic root atherosclerotic plaque [156]. New choline TMA-lyase inhibitors, including iodomethylcholine (IMC) and fluoromethylcholine (FMC), have been recently developed, but data are currently limited [17]. Other studies focused on the suppression of the FMO3 in animal models through an antisense oligonucleotide-based approach, highlighted a marker reduction of diet-enhanced atherosclerosis paralleled by a decrease in TMAO serum levels [157].

### 6.5. Faecal Microbiota Transplantation

Fecal microbiota transplantation (FMT) is an effective therapeutic strategy in multiple gastrointestinal pathologies and consists of transferring a structured community of intestinal bacteria derived from a stool donor in the affected subject. Although the possible implications of FMT in CVD have not yet been investigated in humans, preliminary results are encouraging. In particular, a recent study reported that the gut microbiota of spontaneously hypertensive rats was characterized by increased abundance of *Turicibacter*, which was positively associated with arterial hypertension, and by an altered T helper-17/regulatory T cells (Th17/Tregs) balance in mesenteric lymph nodes. When transplanted in normotensive rats, this dysbiotic microbiota induced endothelial dysfunction and hypertension, through both T cell activation and IL-17 production. Conversely, FMT from normotensive rats in spontaneously hypertensive rats improved systolic blood pressure, endothelial dysfunction, oxidative stress, and vascular inflammation, as well as the imbalance between Th17/Tregs [158].

## 7. Conclusions

Over the last decade, several studies have strengthened the concept of gut microbiota as a dynamic living entity, which can generate, sustain, and worsen various pathological processes, but at the same time influenced by diet, drugs and other stimuli from the external environment. CVD is a global health problem affecting millions of people, and it is of paramount interest to identify effective prevention and treatment strategies to reduce health care costs. Currently, scientific evidence proves the existence of a two-way relationship between the gut microbiota and CVD. The mechanisms involved in this relationship are multiple and extremely complex because they concern immune regulation, inflammatory response, gastrointestinal barrier integrity, metabolic pathways, and much more. Many of these effects are mediated by bacterial-derived products that play a significant role in generating and sustaining chronic inflammation. Human and animal studies have attempted to characterize dysbiosis in CVD, effectively identifying the species most frequently involved, and have analyzed different therapeutic approaches (i.e., dietary intervention, probiotics, prebiotics, drugs, FMT), with encouraging results. Our review sheds light on the complex relationship between intestinal microbiota and CVD by reviewing the latest scientific evidence, focusing on some metabolic diseases. However, further research is needed in order to identify effective microbiome-based preventive and therapeutic approaches to be adopted as additional weapons in the management of CVD.

## Figures and Tables

**Figure 1 ijms-24-09087-f001:**
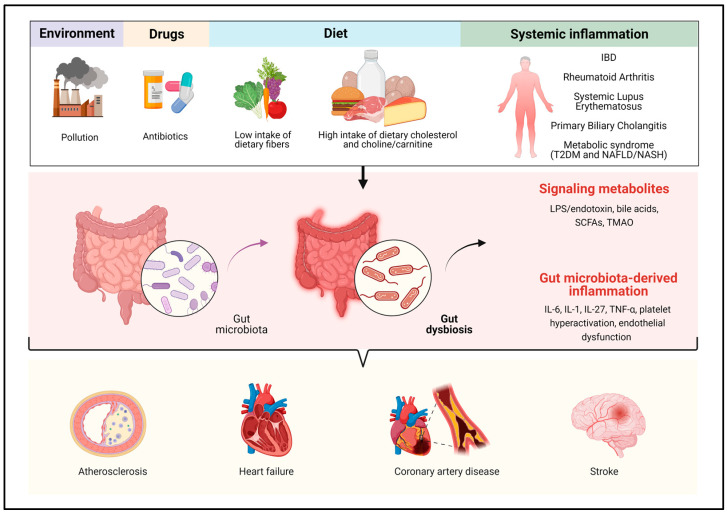
Gut dysbiosis is linked to endogenous and exogenous risk factors, the latter related to several systemic inflammatory and metabolic conditions. Modifications in the gut microbiome composition can lead to alterations of its metabolic pathways and facilitates the translocation of bacteria and their fragments and products in the bloodstream. This can enhance the pro-inflammatory milieu and produce metabolic perturbations that are a fertile ground for cardiovascular disorders. IBD: inflammatory bowel disease; T2DM: type 2 diabetes mellitus; NAFLD: non-alcoholic fatty liver disease; NASH: non-alcoholic steatohepatitis; LPS: lipopolysaccharide; SCFAs: short-chain fatty acids; TMAO: trimethylamine N-oxide; IL-6, IL-1, IL-27: interleukin-6, -1, -27; TNF-α: tumor necrosis factor alpha. Created with Biorender.com ^®^.

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
