# Peer review of "Gut Microbiota and Cardiovascular Disease: Evidence on the Metabolic and Inflammatory Background of a Complex Relationship"

_ijms, 2023, doi:10.3390/ijms24109087_

Round 1
Reviewer 1 Report
This is a valuable contributionto the field of gut microbiota and CVD. There are two topics that woud be appropriate to be included.
1. Influence of gut microbiota on plasma lipoprotein patterns and
2. In the ¶ "drugs", the class of lipid lowering drugs on gut microbiota should be added.
Reviewer 2 Report
This review presents the effect of gut microbiota and its metabolites on the progression of cardiovascular diseases. The manuscript is well presented and organized. However, some remarks and comments need to be answered as followed.
1- In the section 2, the authors present firstly the effects of cigarette smoke on gut microbiota. This section should appear later, not firstly.
2- The authors present the role of inflammation on the gut microbiota in section 3.7. The authors should discuss about the inflammasome NRLP3. This inflammasome mediates the secretion of inflammatory mediators through Caspase-1. It is implicated in the immune system and the gut microbiota, and participates also in the development of atherosclerosis.
3- In the abstract, some abbreviations are defined but not used. The authors should remove them. Examples: Line 22 TMO and SCFA.
4- The authors should use international units such as µmol/L instead of µM.
The english seems to be good.
Reviewer 3 Report
In this manuscript, the authors indicate Gut microbiota and cardiovascular disease: evidence on the metabolic and inflammatory background of a complex relationship. The manuscript can be accepted after addressing the below mentioned corrections.
1. In introduction section, After sentences, ‘Cardiovascular disease (CVD) is one of the leading causes of death and morbidity worldwide.’’ It should be given information. For this purpose the authors can look at the following articles for introduction section: Journal of Biomolecular Structure and Dynamics 40 (1), 77-85 and Archives of physiology and biochemistry 125 (5), 387-395.
2. In introduction section After sentences, ‘In this regard, intensive medical and scientific activity is currently employed in the management of traditional risk factors, but despite maximum medical therapy, there is still a residual risk of undetermined etiology.’’ It should be given information. For this purpose the authors can look at the following articles for this section: Comparative biochemistry and physiology part C: toxicology & pharmacology, 226, 108608 and Journal of Biochemical and Molecular Toxicology 36 (11), e23180
3. In introduction section After sentences, ‘Vitamin K2 or menaquinone (MK) is a vitamin K isoform and a cofactor involved in the carboxylation of several proteins.’’ It should be given information. For this purpose the authors can look at the following articles for this section: Environmental toxicology and pharmacology 70, 103195 and Drug development research 81 (5), 628-636
4. In section 6.1. After sentences Conversely, a high-fat diet leads to unfavourable changes of gut microbiota, faecal metabolomic profile and systemic inflammation.’’ It should be given information. For this purpose the authors can look at the following articles for this section: Journal of Molecular Recognition 36 (3), e3004 and Journal of Molecular Recognition 35 (12), e2987
5. There are several English language issues. It should be corrected.
Minor
Reviewer 4 Report
While the topic is of interest to the general audience, the authors have recycled some previous topics for this review. However, extensive work has not been done and I am not satisfied by the quality. Only one figure, section 6.1/2/3 could benefit from addition of tabulated information.
Moreover, recent reviews on the same topic have not been cited. List is below:
https://www.frontiersin.org/articles/10.3389/fcimb.2022.903570/full
https://www.mdpi.com/2075-1729/12/12/1986
https://microbiomejournal.biomedcentral.com/articles/10.1186/s40168-020-00821-0
Authors do not mention strength or shortcomings of their review. How is it adding to the literature when similar reviews are present on the topic.
Seems fine but some places difficult to understand the meaning.
Reviewer 5 Report
This is an interesting, and relevant, article discussing the contribution of the gut microbiome, to human disease. Specifically, the connection with cardiovascular diseases and the potential for preventing these disorders through manipulation of the microbiome. The concept of dysbiosis in the generation of pathological states is intriguing. The discussion of drugs that positively and negatively affect the gut microbiome was especially interesting.
Line 20-21
Maybe a clarification that the human-microbiome association is the superorganism.
“Indeed, as a metabolically active superorganism, it can affect host physiology”
Lines 74-76
Does this mean the ceasing of tobacco use?
“There is also little evidence about the effect of sustained interruption of tobacco use on the gut microbiota, which seems mainly to produce an increase in Firmicutes and a reduction in Bacteroidetes.”
Line 80-81
Is this correlation negative (as one increases the other decreases) or positive (as one increases, the other also increases)?
“the abundance of the Robinsoniella was negatively associated with systolic blood pressure”
The quality of English used needs some improving.
Round 2
Reviewer 3 Report
The manuscript can be accepted this form.
Reviewer 4 Report
Authors have revised the paper and addressed shortcomings. I recommend publication.